# Spirometry and Smoking Cessation in Primary Care: The ESPIROTAB STUDY, A Randomized Clinical Trial

**DOI:** 10.3390/ijerph192114557

**Published:** 2022-11-06

**Authors:** María del Mar Rodriguez-Alvarez, Josep Roca-Antonio, Silvia Martínez-González, Victoria Vilà-Palau, Carla Chacón, Alexandre Ortega-Roca, Eulàlia Borrell-Thiò, Susana Erazo, Jordi Almirall-Pujol, Pere Torán-Monserrat

**Affiliations:** 1Canet de Mar Primary Care Centre, Catalan Institute of Health (ICS), 08360 Canet de Mar, Spain; 2Unitat de Suport a la Recerca Girona, Fundació Institut Universitari per a la Recerca a l’Atenció Primària de Salut Jordi Gol i Gurina (IDIAP J Gol), 17002 Girona, Spain; 3Department of Medicine, Faculty of Medicine, University of Girona, 17004 Girona, Spain; 4Unitat de Suport a la Recerca Metropolitana Nord, Fundació Institut Universitari per a la Recerca a l’Atenció Primària de Salut Jordi Gol i Gurina (IDIAP J Gol), 08303 Mataro, Spain; 5Llefià Primary Care Center, Catalan Institute of Health (ICS), 08006 Badalona, Spain; 6Santa Coloma De Farners Primary Care Center, Catalan Institute of Health (ICS), 17007 Girona, Spain; 7Mataro 6 (Gatassa) Primary Care Center, Catalan Institute of Health (ICS), 08302 Mataro, Spain; 8Sant Roc Primary Care Center, Catalan Institute of Health (ICS), 08916 Badalona, Spain; 9Cardedeu Primary Care Center, Catalan Institute of Health (ICS), 08440 Cardedeu, Spain; 10Intensive Care Unit, Maresme Health Consortium (CSdM), 08916 Mataro, Spain; 11Germans Trias i Pujol Research Institute (IGTP), 08916 Badalona, Spain; 12Multidisciplinary Research Group in Health and Society, GREMSAS (2017 SGR 917), 08007 Barcelona, Spain

**Keywords:** smoking cessation, clinical trial, primary health care, spirometry

## Abstract

This study aims to evaluate the effect of regularly reporting spirometry results during smoking cessation counseling from a primary care physician on the quit rate in adult smokers. Methods: A randomized, two-arm intervention study was conducted at six primary care centers. A total of 350 smokers, ≥18 years of age, who consulted their primary care physician, participated in the study. At the selection visit, smokers who gave their consent to participate underwent spirometry. Subsequently, an appointment (visit 0) was scheduled to complete a nicotine dependence test, a smoking cessation motivation questionnaire, and a sociodemographic questionnaire. Participants were also offered brief, structured advice on how to quit smoking, as well as detailed information on spirometry results. Patients were then randomized and scheduled for follow-up visits at 3, 6, 12, and 24 months. Both arms received brief, structured advice and detailed information on spirometry results at visit 0. At consecutive follow-up visits, the control group only received brief, structured smoking cessation advice, while the intervention group also received information on initial spirometry results at visits 3 and 6, and a spirometry retest at visit 12. Exhaled carbon monoxide testing was used to check smoking cessation. Results: The study included 350 smokers; 179 were assigned to the control group and 171 to the intervention group. Smoking cessation at one year was 24.0% in the intervention group compared to 16.2% in the control group. At two years, it was 25.2% in the intervention group and 18.4% in the control group. Overall, the adjusted odds of quitting smoking in the intervention group were 42% higher than in the control group (*p* = 0.018). Conclusions: Regular and detailed feedback of spirometry results with smokers increases smoking cessation. Specifically, the likelihood of quitting smoking in the intervention group is 1.42 times higher than in the control group (*p* = 0.018).

## 1. Introduction

Each year, approximately more than 8.7 million people worldwide die from tobacco-related problems. Between 20% and 35% of the years lost to disability and premature death during the productive ages of 45 to 64 years are due to tobacco-attributable diseases [1]. Nevertheless, the World Health Organization (WHO) estimates that, in 2018, there were 1,337,000 smokers worldwide, a figure that will continue to decline given the reduction in consumption, reaching 1,299,000 in 2025. In light of these figures, it will be difficult to reach the goal set by governments to reduce tobacco consumption by 30% by the year 2025 [2].

The efficacy of smoking cessation in reducing the risk of cardiovascular and respiratory diseases surpasses all other interventions, with the benefits being greater the earlier smoking cessation is achieved. Smoking cessation medical counseling (SCMC) remains the initial intervention of choice for treating smokers and is, therefore, recommended as the basis of any intervention strategy. Effectiveness rates of SCMC vary between studies, although the differences found are small and range from 2 to 11%. Stead et al. in their systematic review observed a small but nonsignificant increase in quit rate (RR 1.66, 95% CI 1.42 to 1.94) in favor of SCMC. This roughly equates to an absolute difference in quit rate of about 2.5% [3,4,5].

This small effect on cessation rates may be increased by 1 to 3% with a more intensive intervention. Additional strategies appear to offer further benefits [6,7].

Interventions such as group or individual behavioral therapies (alone or combined with drugs) and pharmacotherapy (bupropion, nicotine substitutes, clonidine, cystine, and varenicline) have been shown to increase quit rates. Efficacy often decreases dramatically at follow-up. Behavioral therapy alone achieves quit rates between 30 and 54%; the combination of two drugs, between 25 and 30%; the combination of drug treatment and behavioral therapy, 37.2% [6,8,9,10,11,12,13,14]. Behavioral therapy with nursing also increases smoking cessation with an RR of 1.29 (95% CI 1.21–1.38) [15].

Other interventions used have not found evidence to support their benefit such as physical exercise, especially cardiovascular, associated with behavioral therapy, hypnotherapy, silver acetate, naltrexone, nicotine shots, lobeline, or nicobrevin [16,17,18,19,20,21,22].

Self-help materials or telephone advice may have a small additional benefit when combined with brief advice [23,24,25].

It is essential when approaching the quitting process to be aware of the five stages of change described by Prochaska and Di Clemente: precontemplation, contemplation, preparation, action, and maintenance. These stages can occur several times in the same smoker, mainly due to relapses. Interventions may have a different impact depending on the stage of change of the smoker [26].

Moreover, 70% of smokers have some form of contact with their general practitioner each year, which means these primary care physicians are in a privileged position to advise and assist patients to quit smoking. These are key opportunities for intervention. Successful smoking cessation has been found to be higher (7.2%) among those who see a doctor than among those who have not seen a doctor in the past year (3.9%) [27].

Most primary care centers in Spain have access to and can perform spirometry, which is indicated to assess abnormal respiratory function in smokers.

Few studies have evaluated the independent impact of spirometry or lung age feedback on smoking cessation and offer opposing results. In other studies, spirometry results are part of a multifactorial intervention, which makes it difficult to assess the isolated impact of the test on smoking cessation. These studies also vary in both the results and the methodology applied. Consequently, there is currently little evidence in the literature that smoking cessation counseling that includes spirometry results increases quit rates [7,27,28].

In this context, we hypothesized that providing patients with information about spirometry results and the impact of smoking on their lung function could play a supportive role in the decision to quit. Our aim was to evaluate the effectiveness of including information on spirometry results by the physician in the SCMC. In doing so, we aimed to define the independent role of informing patients about their spirometry results in smoking cessation strategies.

## 2. Materials and Methods

The present study has been registered on Clinicaltrials.Gov (NCT01296295). In addition, the study methodology, design, and protocol have been published and are available open-access [29].

We conducted a multicenter, randomized clinical trial with an allocation ratio of 1:1. No changes were made to the methods after starting the trial. The study flowchart is available in Figure 1.

The study was carried out at six primary care centers in the province of Barcelona (Spain) with different sociodemographic characteristics. The centers were categorized by setting as rural, semi-rural, or urban. Recruitment was carried out from 2005 to 2008 and follow-up lasted two years, being completed in 2010 after reaching the expected sample size.

### 2.1. Selection Criteria

Inclusion criteria were being an active smoker, being ≥18 years of age, and being seen by the general practitioner for any reason.

Exclusion criteria were having prior diagnosis of COPD, contraindication to spirometry, inability to reach by telephone, communication difficulties (cognitive, sensory, or language impairment), serious disease with poor prognosis (life expectancy less than one year), and refusal to participate in the study.

It was calculated that a sample size of 187 smokers in each group would make a difference of 10% or more detectable between both groups in terms of smoking cessation rate with a reduction in one of the groups, accepting an alpha risk of 0.05 and a beta risk of 0.10. This calculation included a 5% loss during follow-up.

### 2.2. Recruitment

Patients visiting the clinic for any reason were asked about their smoking habits; the study was explained and they were invited to participate. Smokers who gave their consent were asked to perform spirometry followed by a post-bronchodilator test, peak flow test, and pulse oximetry. An appointment was scheduled within one month for the study’s baseline visit. Only patients reporting current tobacco use were included.

### 2.3. Randomization

Participants were then randomly assigned to the control group (CG) or intervention group (IG). The randomization was computer-based, assigning each patient a correlative number from 1 to 500. The computer program assigned a random two-digit number from 00 to 10 to each number: even-numbered patients were assigned to the control group and odd-numbered patients to the intervention group. The randomization process was carried out under blinded conditions, as patients were assigned a correlative number as they were recruited and were subsequently assigned to the control group or intervention group by computer. Therefore, the coordinating center did not know the characteristics of the participants.

### 2.4. Intervention

At the baseline visit, the primary care physician gave all patients brief, structured counseling on quitting smoking, following the recommendations of the Catalan Society of Family and Community Medicine [30]. Moreover, the physician provided structured feedback on the spirometry results. The duration of the visit was approximately 15 min. Lastly, participants completed a questionnaire to collect data on affiliation, sociodemographics, medical history, chronic medication, smoking habits, motivation (Richmond Test), dependency (Fagerstrom Test) [31], cessation phase, and respiratory symptoms.

Therefore, the intervention was the same in both groups at the baseline visit. It differed at follow-up when only the intervention group received reinforced explanation of the spirometry results. Brief smoking cessation counseling was given to both groups at follow-up. At one year of follow-up, only the intervention group repeated the spirometry and received feedback on the results and the impact on their health. Patients who reported quitting smoking underwent a co-oximetry (CO) test.

Spirometries were performed by specially trained nursing staff with a Sibelmed Datospir 120 spirometer. ATS/ERS guidelines [32] and reference values from a Mediterranean population [33] were used. The research team reviewed all the spirometries to assess the acceptability of the test technique and the spirometry pattern.

Patients with an obstructive or mixed pattern (FEV1/FVC post bronchodilator <70) were classified as having a new COPD diagnosis. Severity was classified by FEV1, with 65–79% FEV1 being mild, 50–64% being moderate, 35–49% being severe, and <35% being very severe.

Standardized information based on content defined by the research team was given to patients about their spirometry results depending on their pattern type (Appendix A).

### 2.5. Follow-Up

Follow-up consisted of two telephone consultations and two in-person visits. The telephone consultations were conducted by two nurses trained for this purpose and lasted approximately 10 min. The in-person visits at one and two years were carried out by the same personnel who carried out the baseline visit.

### 2.6. Smoking Cessation

At follow-up, both at one and two years, participants were asked whether they had quit smoking. At the one-year visit, all participants who reported having quit smoking performed the exhaled carbon monoxide test; they were considered nonsmokers if they had a co-oximetry level of <10 ppm. At this visit and the two-year visit, patients who had not smoked for one year or more were considered abstinent smokers, while those who had quit less than a year ago were considered in the process of quitting (quitters).

### 2.7. Analysis Plan

The frequency and percentage of each category were used for the descriptive analysis of the qualitative variables. The median and maximum and minimum values were used for the quantitative variables.

To assess the impact of patients lost to follow-up, an analysis was performed including them and considering that they had not quit smoking. No significant differences were observed between the results of this analysis and the analysis excluding these patients. For this reason, it was decided to present the results of the analysis excluding the missing patients and, therefore, making no assumptions about the data.

As for the bivariate descriptive analysis, both for the comparison of variables between patients lost to follow-up and patients who completed the study, as well as for the comparison of variables between the control and intervention group, the chi-squared test was used when dealing with two qualitative variables (or Fisher’s exact test when the expected frequency for a given cell was less than 5). When dealing with a quantitative and a qualitative variable from two categories, the Student’s t test was used for independent observations to compare the means, and when the qualitative variable had more than two categories, an analysis of variance (ANOVA) was used. To compare medians, the nonparametric test of equality of medians was used.

To assess the effect of the intervention on smoking cessation, univariate logistic regression models were estimated, as well as multivariate models to adjust for possible confounding factors. These analyses were performed with cross-sectional and longitudinal data. In the cross-sectional analyses, a univariate and a multivariate regression model were adjusted with the results of each of the four follow-up visits. This analysis made it possible to estimate the individual effect of the intervention at 3, 6, 12, and 24 months. Given the longitudinal nature of the data (patients “measured” at five different times: at baseline and 3, 6, 12, and 24 months), an overall estimate of the effect of the intervention was made using logistic regression for longitudinal data. To this end, we fit a generalized estimating equations (GEEs) logistic regression model to take into account the correlation of repeated measurements. The dependent variable was smoking cessation and the independent variables were the group (control or intervention) and time of visit (3, 6, 12, and 24 months). Multivariate models were also prepared to minimize the effect of any differences between the two groups.

All comparisons were bilateral and the statistical significance was set at *p* < 0.05. All the analyses were performed with the statistical package R version 3.6.3, supported by the R Core Team and the R Foundation for Statistical Computing, created by statisticians Ross Ihaka and Robert Gentleman, used in Girona (29 February 2020).

## 3. Results

A total of 361 patients participated in the study. Before being randomly assigned to the groups, 11 were excluded due to failure to perform baseline spirometry despite repeated requests. The remaining 350 patients participated in the study; 179 were assigned to the control group and 171 to the intervention group. Figure 1 depicts the flow of patients in the study. Table 1 and Table 2 present the baseline characteristics of participants.

There were no differences between the groups except that the IG comprised more men, started smoking at earlier ages, and consumed fewer anxiolytics (Table 1).

There were no differences between the groups regarding the initial spirometry diagnosis (Table 2). A total of 20.6% of the 350 patients presented fixed airflow obstruction and, as these were previously undiagnosed patients, they were considered new COPD diagnoses, with most having moderate obstruction at the time of diagnosis.

During the two years of follow-up, 32 participants left the study (9.1%): 15 patients from the IG and 17 from the CG (Figure 1). In addition, 30.6% dropped out due to moving, 22.5% due to death, and the remaining 46.9% due to failure to attend follow-up visits without establishing a specific reason. These patients were not different from the 318 patients who completed the 2-year follow-up.

The overall long-term abstinence rate at one year was 21.6%. In the cross-sectional analysis at each visit, smoking cessation was always higher in the intervention group. The difference between the two groups was marginally significant at one and two years. Upon adjustment, these results remained (Table 3).

We analyzed whether smoking cessation was a function of the severity of COPD at the time of entering the study and found that cessation increased slightly as the severity of the COPD increased (e.g., the odds of smoking cessation were 83% higher in patients with severe COPD compared to those with mild COPD), but the difference was not statistically significant (Table 4), nor was the trend test statistically significant (*p* = 0.461, results not shown). When analyzing the effect of our study on the improvement of spirometric parameters, we observed that FEV1 decreased by 1.6% at one year in the CG, while it increased by 2.0% in the IG. At two years, the decrease in the control group was 4.9% and was maintained in the IG. Both at one and two years, there were no statistically significant differences. We also found no differences between those who dropped out and those who did not.

Table 5 presents a longitudinal analysis of data to estimate the overall effect of the intervention regardless of the time of the visit. In the intervention group, the odds of smoking cessation were 36% higher than in the control group, which is statistically significant (*p* = 0.029). This increase is maintained after adjusting for differences between the two groups (42%, *p* = 0.018).

When the statistical analysis was repeated to include the 32 patients who dropped out of the study and was considered as continuing to smoke, the difference in smoking cessation in the intervention group continued to be statistically significant.

One variable that could influence or alter the final results of our study would be pharmacological treatment. In our intervention, no pharmacological treatment was offered to quit smoking. The only intervention carried out was that summarized in the study protocol. However, during follow-up, no weight was given to the habitual practice of each doctor and, therefore, patients may have received pharmacological treatment. When analyzing this variable, we observed no significant differences between both groups. No differences were observed in the percentage of quitters or ex-smokers depending on whether or not they had received pharmacological treatment. Even so, the analysis conducted in this aspect was very limited given that only 13 cases used pharmacological treatment.

## 4. Discussion

Smoking cessation is a priority for health systems. Various types of strategies have been explored based on interventions including brief or intensive counseling or information on biomedical risks.

Our intervention consisted of associating brief counseling with detailed and standardized feedback on spirometry results in smokers without previously diagnosed chronic airflow obstruction. This intervention was carried out periodically.

Providing smokers with the results of their lung function as obtained from spirometry in a structured and periodic way increased the odds of quitting by 42%.

The overall rate of abstinence (21.6%) is higher than that found in longitudinal studies performed in Spain and southern Europe, where an annual incidence of smoking cessation ranging from 0.5 to 5% has been observed in different periods [34,35].

Additionally, if we take into account the data from the literature, and assuming an 11% improvement of the intervention efficacy can be attributed to SCMC, that means there is an additional benefit of more than 4% from our intervention [3,4,5,6]. Therefore, it seems logical to conclude that this can be directly attributed to our intervention. This conclusion is strengthened by the fact that the two groups were comparable throughout the study, and assignment to the intervention or control group was completely random.

When comparing our study with the existing literature, we see that the findings are diverse [7,27,36,37,38,39,40,41,42,43,44,45,46,47]. Few studies have assessed the effect of feedback on spirometry results [36,37,38] or lung age [39,40,41,42,43] independently, and they offer opposing results. The dropout rates of previous studies range from 2 to 24% in the control group and 6.5 to 32% in the intervention group, with a higher dropout rate in the IG, although it is not statistically significant [37,38,39,41].

However, in contrast to ours, the study population in these other studies presented high motivation and a high rate of pharmacological treatment to aid smoking cessation [38,42], feedback that was associated with a more intensive intervention [39], a very small sample size [41,43], specific characteristics such as being drug users [41], and a very short follow-up period [42], which makes it difficult to find significant differences.

Only three studies found statistically significant differences [36,40,42], two of them regarding feedback on lung age results [40,42].

One study with characteristics similar to ours is by Martin-Lujan et al. [36], who observed significant cessation rates at 12 months in favor of the intervention group (5.6% vs. 2.1%), although at a lower rate than in our study. In their study, only normal spirometries were randomized, which may have led to a lower cessation rate.

Parker et al. [40] obtained similar results (13.6% IG vs. 6.4% CG as compared to 25.2% IG vs. 18.4% CG in our study), although the follow-up and the message transmitted during the intervention were different.

Segnan et al. [39] repeated the intervention, as we did, at 3, 6, and 9 months, and found no differences between the groups. There were low attendance rates at reinforcement sessions in their study, which could explain the differences with our results. While most studies performed a specific intervention that was not repeated at follow-up [37], in our study, the intervention was repeated at 3, 6, and 12 months, which could explain why our intervention managed to increase and maintain cessation rates.

In studies that assessed the success of associating the reporting of spirometry results with another type of more intensive intervention or carbon monoxide testing, dropout rates varied from 9 to 52.4% in the control groups and from 6.7 to 50.8% in the intervention groups [44,45,46,47,48]. The population characteristics of these studies were quite similar to ours. Sippel et al. [47] used a different message depending on whether the spirometry result was normal or obstructive, just as we did.

In these studies, it is difficult to assess the isolated impact of spirometry because its effect forms part of an intense motivational intervention. Moreover, there are differences in the limitations and methodologies between these studies and ours, which makes it difficult to compare them and means we must do so with caution. Still, in most of these studies, the population characteristics were quite similar to ours.

We found a prevalence of COPD in previously undiagnosed smokers of 20.6%, which is similar to previous findings [36,37,44].

Several studies analyzed whether more patients with chronic airflow obstruction quit smoking than those with normal spirometry patterns, although here, too, the results are contradictory [44,49,50,51]. The differences between the studies could be explained by the association or not of pharmacological treatment or exclusion of the most severe COPD patients as a greater severity has been associated with a greater likelihood of smoking cessation [49]. We observed that COPD patients with a moderate-severe obstruction have higher smoking cessation rates than patients with mild-moderate obstruction, but this increase is not statistically significant. Other studies have also detected this relationship [50,51]. However, it must be taken into account that the spirometry values used in these studies differ from ours.

The beneficial effects of smoking cessation can be observed in the short term, as demonstrated in Darabseh’s study, where 14 days after smoking cessation, skeletal muscle fatigue resistance and circulating markers of inflammation were improved, but without significant changes in spirometric values [52]. A reduction in lung inflammation, measured as a reduction in the size and number of pulmonary nodules and improvement in lung function measured as an increase in FEV1, increase in arterial oxygen pressure, and decrease in heart rate, has also been reported at 3 months [53]. In our study, although we observed a greater decrease in FEV1 in the CG than in the IG, these data were not statistically significant.

### Limitations

Due to the type of intervention performed, it was not possible for our study to be a double-blind trial. To prevent bias, the doctors and nurses that participated in this study received prior training on conveying standardized messages.

In the study, no other smoking cessation interventions were conducted, but we did not interfere with the regular practice of each doctor. This means that patients may have been referred to a clinic specialized in smoking cessation, received medical treatment, and/or had other tests performed (repeat spirometry, etc.) if their doctor deemed it necessary. These data were not collected for subsequent analysis, except in 112 cases (32%) of the sample, in which data on whether or not they had received pharmacological treatment were available. In any case, random assignment would have distributed these nonstudy interventions equally between both groups.

## 5. Conclusions

Giving periodic, detailed feedback on spirometry results to smokers increases smoking cessation. Specifically, the odds of quitting smoking in the intervention group are 1.42 times higher than in the controls group (*p* = 0.018). This effect is maintained at 24 months.

## Figures and Tables

**Figure 1 ijerph-19-14557-f001:**
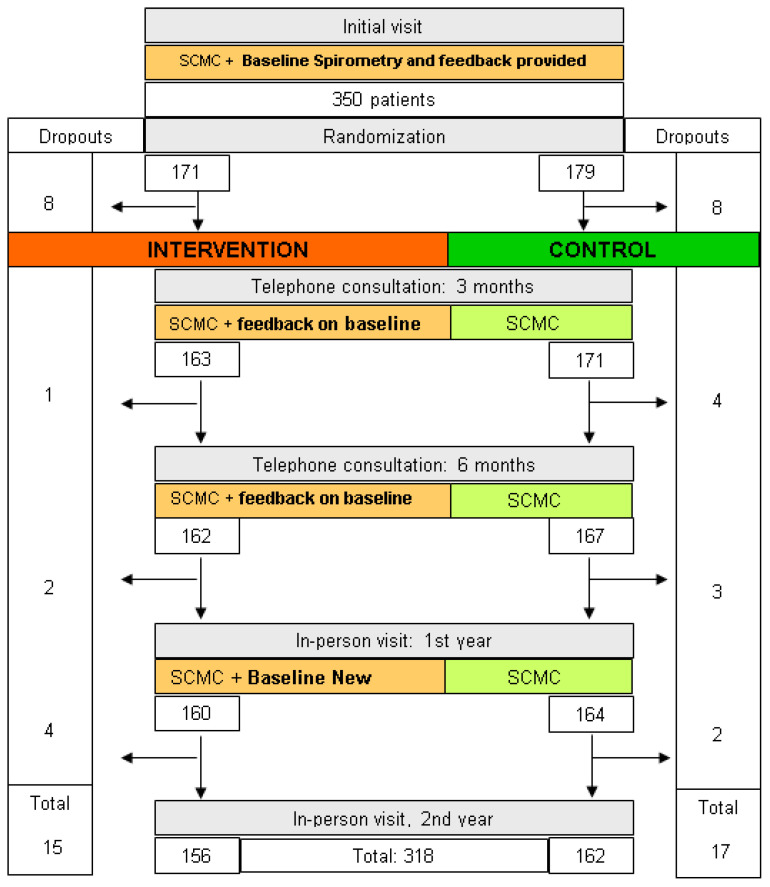
Patient flow and tests performed at the visits.

**Table 1 ijerph-19-14557-t001:** Characteristics of participants at baseline, according to randomized group assignment.

	Control (N = 179)	Intervention (N = 171)	*p*-Value
**Sex female** N (%)	71 (39.7%)	46 (26.9%)	0.016
**Age** (Med. (min-max))	53 (18–80)	51 (19–85)	0.375
**Residence**			
Rural N (%)	17 (9.6%)	15 (8.82%)	0.568
Urban N (%)	102 (57.6%)	90 (52.94%)	
Semi-rural N (%)	58 (32.8%)	65 (38.24%)	
**Education level**			
Illiterate N (%)	14 (8.86%)	7 (4.58%)	0.331
Primary school N (%)	95 (60.13%)	95 (62.09%)	
Secondary school or VE N (%)	33 (20.89%)	41 (26.80%)	
University degree N (%)	9 (5.70%)	5 (3.27%)	
Licentiate degree N (%)	7 (4.43%)	5 (3.27%)	
**Employment status**			
Active N (%)	102 (59.65%)	106 (64.24%)	0.687
Unemployed N (%)	16 (9.36%)	12 (7.27%)	
Retired N (%)	42 (24.56%)	40 (24.24%)	
Housewife N (%)	11 (6.43%)	7 (4.24%)	
**Alcohol consumption** N (%)	75 (42.6%)	84 (50.9%)	0.125
**Drug consumption** N (%)	8 (4.47%)	4 (2.37%)	0.291
**Anxiolytic consumption** N (%)	35 (19.7%)	20 (12%)	0.053
**Antidepressant consumption** N (%)	21 (11.8%)	13 (7.78%)	0.214
**Analgesic consumption** N (%)	24 (13.5%)	22 (13.2%)	0.933
**Cough** N (%)	63 (36.4%)	58 (34.9%)	0.777
Wheezing N (%)	59 (34.1%)	42 (25.3%)	0.077
**Expectoration (sputum)** N (%)	48 (27.7%)	46 (27.7%)	0.994
**Dyspnea** N (%)	30 (17.3%)	24 (14.5%)	0.469
**Starting age** (Med. (min-max))	17 (7–52)	16 (6–55)	0.012
**No. cigarettes day**			
<10 N (%)	24 (13.6%)	20 (11.8%)	0.656
10–29 N (%)	112 (63.3%)	103 (60.9%)	
≥30 N (%)	41 (23.2%)	46 (27.2%)	
**No. packs year** (Med. (min-max))	28 (0.6–189)	30 (1–129)	0.537
**No. quit attempts**			
0 N (%)	67 (59.8%)	63 (60.6%)	0.975
1–2 N (%)	24 (21.4%)	21 (20.2%)	
>2 N (%)	21 (18.8%)	20 (19.2%)	
**Motivation test**			
Low N (%)	52 (45.2%)	40 (37.4%)	0.197
Moderate N (%)	28 (24.3%)	22 (20.6%)	
High N (%)	35 (30.4%)	45 (42.1%)	
**Dependence test**			
Low N (%)	27 (20.6%)	24 (21.6%)	0.892
Moderate N (%)	76 (58.0%)	66 (59.5%)	
High N (%)	28 (21.4%)	21 (18.9%)	

Med. (min-max): Median (minimum—maximum). VE: Vocational education.

**Table 2 ijerph-19-14557-t002:** Spirometry diagnosis at study intake: comparison of the two groups.

	Total (N = 350)	Control (N = 179)	Intervention (N = 171)	*p*-Value
**Initial spirometry diagnosis**				
Normal	125 (35.7%)	65 (36.3%)	60 (35.1%)	0.149
Obstructive	29 (8.3%)	14 (7.8%)	15 (8.8%)	
Non-obstructive	99 (28.3%)	48 (26.8%)	51 (29.8%)	
Mixed	43 (12.3%)	17 (9.5%)	26 (15.2%)	
Small airways	54 (15.4%)	35 (19.6%)	19 (11.1%)	
**COPD** (*)				
Mild	16 (22.5%)	7 (22.6%)	9 (22.5%)	0.962
Moderate	40 (56.3%)	17 (54.8%)	23 (57.5%)	
Severe	15 (21.1%)	7 (22.6%)	8 (20.0%)	
Very severe	0 (0.0%)	0 (0.0%)	0 (0.0%)	
Pharmacologic treatment (**)	13 (3.7%)	7 (3.9%)	6 (3.5%)	0.842

The number of patients (%) is shown. (*) Only the 72 patients with an obstructive or mixed pattern. COPD could not be categorized in one case in the intervention *group.* (**) The percentage of patients who used pharmacological treatment during the quitting process is collected. The treatment used was nicotine replacement therapy, bupropion, or varenicline.

**Table 3 ijerph-19-14557-t003:** Cross-sectional logistic regression analysis of the effect of the intervention on smoking cessation at each visit.

	Control (N = 179)	Intervention (N = 171)	OR (*) (95% CI), *p*-Value	
			Unadjusted	Adjusted (**)
**Quitters (***)**				
3 months	27 (15.1%)	27 (15.8%)	1.06 (0.59–1.89), 0.855	1.00 (0.54–1.87), 0.995
6 months	28 (15.6%)	34 (20.0%)	1.35 (0.78–2.35), 0.288	1.35 (0.75–2.44), 0.314
12 months	29 (16.2%)	41 (24.0%)	1.63 (0.96–2.79), 0.071	1.73 (0.98–3.09), 0.060
24 months	33 (18.4%)	43 (25.2%)	1.49 (0.89–2.49), 0.129	1.63 (0.94–2.86), 0.084
**Ex-smoker (****)**				
12 months	17 (9.5%)	23 (13.5%)	1.56 (0.81–3.08), 0.188	1.45 (0.72–3.01), 0.304
24 months	23 (12.9%)	34 (19.9%)	1.68 (0.95–3.03), 0.077	1.66 (0.90–3.11), 0.108

The number of patients who quit smoking and their percentage out of the total are shown. (*) OR: Compares the odds of being a quitter or ex-smoker in the intervention group compared to the control group. (**) Adjusted for age, sex, age at which the patient began smoking, and use of anxiolytics. (***) Patients who quit smoking at some point in the study even if they later relapsed were considered to be quitters. (****) Patients who had not smoked for more than one year were considered to be ex-smokers.

**Table 4 ijerph-19-14557-t004:** Smoking cessation in COPD patients by severity (N = 72).

		Smoking Cessation		
		Yes	No	OR (95% CI) (***), *p*-Value
COPD (*)	Mild **	3 (21.4%)	11 (78.6%)	1
	Moderate **	9 (24.3%)	28 (75.7%)	1.18 (0.29–6.05), 0.828
	Severe	5 (33.3%)	10 (66.7%)	1.83 (0.36–10.89), 0.476

(*) In one COPD case, it was impossible to classify severity. (**) In two cases with mild and three with moderate COPD, it was impossible to certify smoking status. (***) OR: Compares the odds of smoking cessation in cases of moderate and severe COPD with cases of mild COPD.

**Table 5 ijerph-19-14557-t005:** Longitudinal logistic regression analysis of the effect of the intervention on smoking cessation.

	Dependent Variable: Smoking Cessation	
	Unadjusted	Adjusted (*)
	OR (*) (95% CI), *p*-Value	OR (*) (95% CI), *p*-Value
IG vs. CG	1.36 (1.03–1.80), 0.029	1.42 (1.06–1.90), 0.018
Visit		
3 months	1	1
6 months	1.23 (0.81–1.85), 0.337	1.23 (0.81–1.86), 0.333
12 months	1.37 (0.92–2.06), 0.125	1.38 (0.92–2.08), 0.121
24 months	1.51 (1.01–2.25), 0.045	1.52 (1.01–2.28), 0.043

(*) Adjusted for age, sex, age at which the patient began smoking, and use of anxiolytics. The adjusted model did not include the interaction between time of visit and group, as it was not statistically significant.

## Data Availability

Prior presentations: The protocol of this study was published in BMC Family Practice (2011) and the preliminary results were published in Primary Care Respiratory Journal (2008) as they were the subject of an oral report at the International Primary Care Respiratory Group World Conference (IPCRG, Seville, Spain 2011) and the 17th WONCA Europe Conference (Warsaw, Poland 2011). This study has also been the subject of the Doctoral Thesis: Effect of smoking cessation counseling combined with a detailed discussion of spirometry results by primary care physicians on smoking in adult smokers, supervised by Dr. Jordi Almirall i Pujol. Universitat Autònoma de Barcelona (2013). ddd.uab.cat/pub/tesis/2013/hdl_10803_117377/1 accessed 7 February 2021.

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
