# Peer review of "Spirometry and Smoking Cessation in Primary Care: The ESPIROTAB STUDY, A Randomized Clinical Trial"

_ijerph, 2022, doi:10.3390/ijerph192114557_

Round 1

Reviewer 1 Report

Overall, I think this is well written, and the first paper I have read that has looked at explanation of results as the intervention. I was not sure why the spirometry results were explained to the control group at the initial appointment as this muddies the results a little. The other intervention that I would like to see explained more in the paper is the use of pharmacologic agents. If possible, can you list which medications are used as this may differ from different parts of the world. I would also like to see this added to Table 1. 

Additional comments:

Figure 1: clarify baseline and feedback so figure can standalone

Paragraph 2 "This intervention was carried out periodically." I do not know what this means, please revise.

Background and Discussion cite different studies for success rates of SCMC. Please revise.

Several commas are used in the place of decimals for numbers. Please fix.

Reviewer 2 Report

Abstract

Line 34: Authors should remove the word ‘’over’’ in (smokers over the age of ≥18) since this will be repetition with the symbol ≥ (greater than or equal to).

Introduction:

Add about the (5 A’s) 5 stages of quitting in SC.

Add about behavioural support and modes used in SC (pharmacological treatment, aerobic exercise etc )

Add and discuss the acute (short-term) effect of SC on spirometry (see: Fourteen days of smoking cessation improves muscle fatigue resistance and reverses markers of systemic inflammation)

Methods:

2.1 Selection criteria

Were the smokers included in the study dual users (shisha-waterpipe smokers, vapers at the beginning of vape era ..) this would make a big difference.

Sample size calculation: please explain more.

Have you followed blindness in the randomization process? If so, add and if not why? (beside what mentioned in the limitations).

Spirometry technique: which guidelines were followed for conducting the spirometry? Was it ATS/ERS guidelines? If so, add a reference and a sentence mentioning that.

Results and discussion:

A clear question is, have you looked at any enhancement of the respiratory function in your participants throughout the study period?

I believe this would be a great add on to the study.
